



# Fossil fuel combustion, biomass burning and biogenic sources of fine carbonaceous aerosol in the Carpathian Basin

Imre SALMA[1], Anikó VASANITS-ZSIGRAI[1], Attila MACHON[2], Tamás VARGA[3],
István MAJOR[3], Virág GERGELY[3], Mihály MOLNÁR[3]

[1] Institute of Chemistry, Eötvös University, Budapest, Hungary
[2] Air Quality Reference Center, Hungarian Meteorological Service, Budapest, Hungary
[3] Isotope Climatology and Environmental Research Centre, Institute for Nuclear Research, Debrecen, Hungary

*Correspondence to*: Imre Salma (salma@chem.elte.hu)

**Abstract.** Fine-fraction aerosol samples were collected, air pollutants and meteorological properties were measured in-situ in regional background environment of the Carpathian Basin, a suburban area and central part of its largest city, Budapest in each season for 1 year-long time interval. The samples were analysed for $PM_{2.5}$ mass, organic carbon (OC), elemental carbon (EC), water-soluble OC (WSOC), radiocarbon, levoglucosan (LVG) and its stereoisomers, and some chemical elements. Carbonaceous aerosol species made up 36% of the $PM_{2.5}$ mass with a modest seasonal variation and with a slightly increasing tendency from the regional background to the city centre (from 32 to 39%). Coupled radiocarbon-LVG marker method was applied to apportion the total carbon (TC=OC+EC) into contributions of EC and OC from fossil fuel (FF) combustion ($EC_{FF}$ and $OC_{FF}$, respectively), EC and OC from biomass burning (BB) ($EC_{BB}$ and $OC_{BB}$, respectively) and OC from biogenic sources ($OC_{BIO}$). Fossil fuel combustion showed rather constant daily or seasonal mean contributions (of 35%) to the TC in the whole year in all atmospheric environments, while the daily contributions of BB and biogenic sources changed radically (from <2 up to 70–85%) over the seasons at all locations. In autumn, the three major sources contributed equally to the TC in all environments. In winter, it was the BB that was the major source with a share of 70% at all sites. The contributions from biogenic sources in winter were the smallest, although they were still non-negligible with an increasing share (from 5 to 8%) from the regional background to the urban sites. In spring, FF combustion and biogenic sources were the largest two contributors at all locations with typical shares of 45–50% each. In summer, biogenic sources became the major source type with a monotonically increasing tendency (from 56 to 72%) from the city centre to the regional background. The share of BB was hardly quantifiable in summer. The $EC_{FF}$ made up more than 90% of EC in spring and summer, while in autumn and winter, the contributions of $EC_{BB}$ were considerable. Biomass burning in winter and autumn offers the largest and considerable potentials for improving the air quality in cities as well as in rural areas of the Carpathian Basin.





## 1 Introduction and objectives

Carbonaceous aerosol constituents make up a major part (e.g. 20–60% in the continental mid-latitudes and up to 90% in tropical forests) of the $PM_{2.5}$ mass (Kanakidou et al., 2005; Fuzzi et al., 2015). Their largest emission or production source types are fossil fuel (FF) combustion, biomass burning (BB) and biogenic sources (Le Quéré et al., 2018). These processes also represent the highest source of certain important aerosol species such as soot and of some pollutant or greenhouse gases such as CO, $NO_x$, $CO_2$ and volatile organic compounds (VOCs) on global scale (Wiedinmyer et al., 2011; Tian et al., 2016). The sources produce both primary and secondary particles, and they are linked directly or indirectly to a variety of anthropogenic activities in many ways (Hallquist et al., 2009). The perturbations in atmospheric concentrations and chemical, physical and meteorological properties caused by these sources have important consequences on the Earth system. They include the radiation balance (Lohmann et al., 2000), cloud formation/properties, water cycling and other biogeochemical cycles (Andreae and Rosenfeld, 2008; Cecchini et al., 2017), atmospheric chemistry and nucleation (Fuzzi et al., 2015; Nozière et al., 2015; Kirkby et al., 2016), atmospheric transport/mixing (Rosenfeld et al., 2019), forest growth and agriculture production (Artaxo et al., 2009; Rap et al., 2015), ecosystems (Cirino et al., 2014), built environment and cultural heritage (Bonazza et al., 2005), and human health/wellbeing (Lelieveld and Pöschl, 2017; Burnett et al., 2018). Some particular sources, e.g. fuel wood or agricultural residue burnings are expected to be increased due to their role in decentralised and substitute energy production (Vicente and Alves, 2018). At the same time, their potential disadvantages and risk have been less recognised (Hays et al., 2003; Chen et al., 2017). It is, therefore, highly relevant to estimate the relative contribution of FF combustion, BB and biogenic sources to major carbonaceous aerosol species, namely to organic carbon (OC) and elemental carbon (EC).

Huge number, composite character, spatial and temporal variability of the sources together with the complex mixture and atmospheric transformation of their products make the quantification of these source types or their inventory-based source assessment challenging (Nozière et al., 2015). There are several methods to apportion the particulate matter (PM) mass or carbonaceous species among some or all major source types. They include source-specific marker methods (Fraser et al., 2000; Szidat et al., 2006, 2009; Minguillón et al., 2011; Zhang et al., 2012; Bernardoni et al., 2013), multi-wavelength optical methods (Sandradewi et al., 2008a, 2008b; Zotter et al., 2017; Forello et al., 2019) and various multivariate statistical



methods based on online or offline data (Hopke, 2016; Maenhaut et al., 2016).The latter
approaches are recently also combined with dedicated molecular tracers/fragments and mass
spectrometry or advanced optical techniques (Stefenelli et al., 2019). The marker methods are
advantageous from the point of view that they do not demand many samples or very extensive
data sets and that the required analytical data are ordinarily available in related studies. Among
the most frequently and successfully adopted two markers are radiocarbon ($^{14}$C, $T_{1/2}$=5730 y),
which is used for quantifying FF combustion and levoglucosan (LVG, monosaccharide
anhydride $C_6H_{10}O_5$), which is utilised for assessing BB. The latter molecule is often applied
together with its stereoisomers mannosan (MAN) and galactosan (GAN) since their
concentration ratios were connected to biomass type (e.g. hardwood or softwood; Fine et al.,
2004; Schmidl et al., 2008). Formation, modelling utilisation, atmospheric processes and
analytical determinations of these markers together with their advantages and limitations were
described, evaluated and discussed in detail earlier (e.g. Simoneit et al., 1999, 2004; Fraser and
Lakshmanan, 2000; Nolte et al., 2001; Pashynska et al., 2002; Zdráhal et al., 2002; Puxbaum
et al., 2007; Saarikoski et al., 2008; Caseiro et al., 2009; Fabbri et al., 2009; Szidat et al., 2006,
2009; Favez et al., 2010; Hennigan et al., 2010; Hoffmann et al., 2010; Kourtchev et al., 2011;
Piazzalunga et al., 2011; Maenhaut et al., 2012, 2016; Yttri et al., 2014). The coupled
radiocarbon-LVG marker method, introduced recently (Salma et al., 2017), is a combination
of the two approaches and it allows to apportion the TC (TC=OC+EC) among all major source
types, thus among the contributions of EC and OC from FF combustion (EC$_{FF}$ and OC$_{FF}$,
respectively), EC and OC from BB (EC$_{BB}$ and OC$_{BB}$, respectively), and OC from biogenic
sources (OC$_{BIO}$).

Water-soluble OC (WSOC) is also an important carbonaceous aerosol species because it is
considered as an indicator of secondary organic aerosol (SOA) or carbonaceous particles after
atmospheric chemical aging (Claeys et al., 2010). It is related to more oxygenated chemical
species than freshly emitted or formed organic constituents, and this class of molecules is
expected to contribute substantially to cloud condensation nuclei (CCN) activity of particles
and represent potentially larger negative health effects of particulate mass deposited in the
human respiratory system due to its solubility (Hallquist et al., 2009; Fuzzi et al., 2015; Nozière
et al., 2015).

Despite their overall role together with the health, climate and environmental effects, there are
serious gaps in our knowledge on FF combustion, BB and biogenic sources – particularly on



the latter type in more polluted or urban areas. Information on the properties of the major
apportioned or secondary carbonaceous aerosol species and on their relationships with other
atmospheric quantities have been missing internationally on extended spatial scales as well as
on larger cities. The Carpathian Basin (also known as the Pannonian Basin) is the largest,
topographically well separated, orogenic basin in Europe (Salma et al., 2016b). Its land is
mostly used for intensive agriculture and farming, while larger forested areas with deciduous,
coniferous or mixed wood occur in the inner and bounding mountains. Weather situations
within the basin are generally uniform, which makes it advantageous for studying atmospheric
phenomena and processes. Budapest with 2.3 million inhabitants in the metropolitan area and
with its central geographical location is the largest and principal city in the basin. The mean
green space intensity – which indicates the healthy green coverage – for Budapest in 2015 was
estimated from Landsat satellite images to be approximately 50% with spatial variations from
19% in the city centre to 55% in the suburban zone (Tatai et al., 2017).

As part of a research project, we collected aerosol samples in the regional background
atmospheric environment of the Carpathian Basin, suburban area and city centre of Budapest
in each season for 1 year-long time interval and analysed them for various aerosol constituents,
which are required for source apportionment. The analytical results were complemented by
supporting air pollutant and meteorological data as well. The major objectives of the present
paper are to report the main findings of this research, to discuss the properties and contributions
of FF combustion, BB and biogenic sources and related atmospheric processes, to interpret the
relationships among various variables for different environmental types and seasons, and to
formulate some general conclusions on air quality of the region and city.
**2 Methods**
**2.1 Collection of aerosol samples and in-situ measurements**
The aerosol samples were collected at three sites in Hungary, in a rural background area and at
two urban sites in Budapest (Fig. 1). The samplings at the rural location were realised at K-
puszta station (N 46° 57' 56'', E 19° 32' 42'', 125 m above mean sea level, a.s.l.), which is
situated on the Great Hungarian Plain in a clearing within a mixed forest of coniferous and
deciduous trees (Salma et al., 2016a). The station is part of the European monitoring and
evaluation of the long-range transmission of air pollutants programme (EMEP network) and
represents the largest/main area (regional background) of the Carpathian Basin. One of the



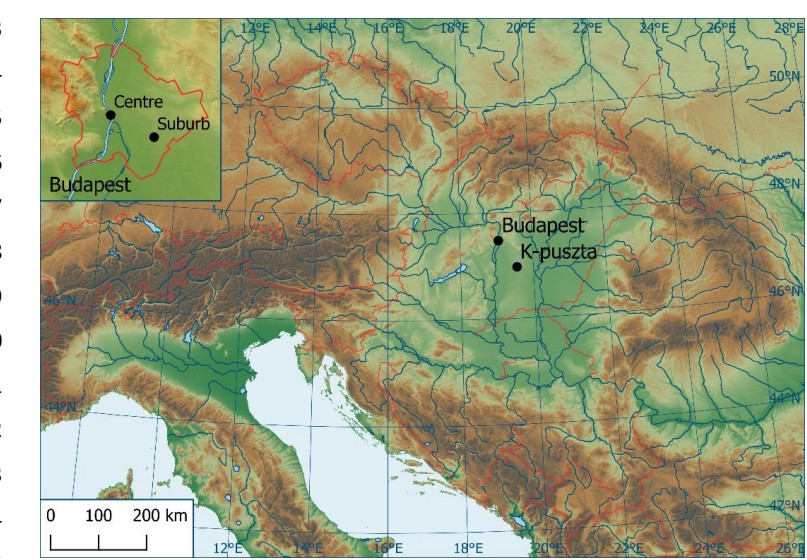

**Figure 1.** The Carpathian Basin with location of the sampling sites in Budapest (city centre and
suburban area) and at K-puszta station (regional background).

urban sites was at an open suburban area of residential Budapest at the Marczell György Main
Observatory of the Hungarian Meteorological Service (N 47° 25' 46'', E 19°10' 54'', 138 m
a.s.l.). The collections at the other urban location were performed at the Budapest platform for
Aerosol Research and Training (BpART) Laboratory of the Eötvös University (N 47° 28' 30'',
E 19° 03' 45'', 115 m a.s.l.). The latter site is situated on the bank of the Danube and represents
a well-mixed average atmosphere of the city centre (Salma et al., 2016a). Some details of the
sampling campaign are summarised in Table 1.

The aerosol sampling was realised by three identical high-volume DHA-80 devices equipped
with $PM_{2.5}$ inlets (Digitel, Switzerland). The collection substrates were quartz fibre filters with
a diameter of 150 mm (QR-100, Advantec, Japan). Daily aerosol samples were taken starting
at 00:00 LT (LT=UTC+1 or daylight-saving time UTC+2). The sampled air volumes were ca.
720 $m^3$. One field blank sample was also taken at each site and season. All filters were pre-
heated at 500 °C for 24 h before the exposure and were stored in a freezer after the collections.





**Table 1.** Start and end dates of the sampling periods and number of aerosol samples collected in regional
background of the Carpathian Basin, suburban area and city centre of Budapest in different seasons.
Initials of the consecutive months allocated to the seasons are indicated in brackets.

| Site type | Season/ Sample | Autumn (SON) | Winter (DJF) | Spring (MAM) | Summer (JJA) |
|---|---|---|---|---|---|
| Region | Period | 18–31. 10. 2017 | 09–22. 01. 2018 | 17–30. 04. 2018 | 17–30. 07. 2018 |
| | Number | 14 | 14 | 14 | 14 |
| Suburb | Period | 18–31. 10. 2017 | 06–22. 01. 2018 | 17–30. 04. 2018 | 17. 07–01. 08. 2018 |
| | Number | 14 | 17 | 14 | 14 |
| Centre | Period | 18–27. 10. 2017 | 10–16. 01. 2018 | 17–23. 04. 2018 | 17–23. 07. 2018 |
| | Number | 7 | 7 | 7 | 7 |


Total particle number concentrations ($N_{6–1000}$) were derived from a differential mobility particle
sizer (DMPS) system with a time resolution of 8 min (Salma et al., 2019). The DMPS
measurements have been performed continuously and according to the recommendations of
international technical standards. Concentrations of criteria air pollutants, i.e. $SO_2$, $NO/NO_x$,
CO, $O_3$ and $PM_{10}$ mass were obtained from regular stations of the National Air Quality
Network. For the regional background and suburban area, they were measured directly at the
sampling sites, while for the city centre, the pollutants were recorded in a distance of 4.5 km
in the upwind prevailing direction from the sampling site. The concentrations are measured by
UV fluorescence (Ysselbach 43C), chemiluminescence (Thermo 42C), IR absorption (Thermo
48i), UV absorption (Ysselbach 49C) and beta-ray attenuation methods (Thermo FH62-I-R),
respectively with a time resolution of 1 h. Local meteorological data including air temperature
($T$), relative humidity (RH), wind speed (WS) and global solar electromagnetic radiation
(GRad) were acquired by standardised meteorological methods at the sites with a time
resolution of 10 min. According to our knowledge, there were no extensive agricultural burns
or wild fires in the basin during to the actual sampling time intervals, and the BB in the area is
expected to be dominated by biofuel utilisation.
**2.2 Analysis of aerosol samples**
The PM mass was determined by weighing each filter before and after the sampling on a
microbalance with a sensitivity of 1 μg. The exposed and blank filters were pre-equilibrated
before weighing at a $T$ of 19–21°C and RH of 45–50% for at least 48 hours. The measured
mass data for the exposed filters were corrected for the field blank values. A few PM mass data



were below the limit of quantitation (LOQ), which was approximately 6 µg m$^{-3}$ or above but
very close to it.

One or two punches with an area of 1.0 or 1.5 cm$^2$ each of the filters were directly analysed by
thermal-optical transmission (TOT) method (Birch and Cary, 1996) using a laboratory OC/EC
analyser (Sunset Laboratory, USA) adopting the EUSAAR2 thermal protocol. The measured
OC data for the exposed filters were corrected for the field blank values, while the EC on the
blanks was negligible. All measured OC and EC data were above the LOQ, which was
approximately 0.09 µg m$^{-3}$.

One or two sections with an area of 2.5 cm$^2$ each of the filters were extracted in water, the
extracts were filtered, and the filtrates were analysed for WSOC by a Vario TOC cube analyser
(Elementar, Germany) in three repetitions with an injected volume of 1 ml each. The measured
WSOC data for the exposed filters were corrected for the field blank values. All measured
WSOC data were above the LOQ, which was ca. 0.08 µg m$^{-3}$.

A section with an area of 2 cm$^2$ of each filter was analysed for LVG, MAN and GAN by gas
chromatography/mass spectrometry (GC/MS) after trimethylsilylation (Blumberger et al.,
2019). The filter sections were extracted repeatedly by dichloromethane-methanol in an
ultrasonic bath. The extracts were filtered and spiked with an internal standard (IS) of methyl
$\beta$-L-arabinopyranoside. The trimethylsilylation was realised by hexamethyldisilazane as
silylating agent, pyridine as solvent and trifluoroacetic acid as catalyst at 70 °C. The prepared
samples were analysed by a Varian 4000 GC-MS/MS system (USA) with a GC/MS column of
SGE forte BPX-5 capillary (length×inner diameter 15 m×0.25 mm; film thickness 0.25 µm,
SGE, Australia). The quantification was carried out in the selected ion monitoring mode by
quantifier ions with mass-to-charge ratios of $m/z$=204 for LVG and of 217 for MAN, GAN and
IS. The LVG data for the exposed filters were corrected for blank values; while MAN and
GAN were not detected in the blanks. The LOQ for LVG and MAN was approximately 1.2 ng
m$^{-3}$, while it was approximately 0.5 ng m$^{-3}$ for GAN. All LVG data were above the LOQ, while
the MAN and GAN could not be quantified in the summer samples.

Filters collected in parallel on seven overlapping days in each season were subjected to C
isotope analysis of the TC content by accelerator mass spectrometry (AMS) with an off-line
combustion system (Molnár et al., 2013; Janovics et al., 2018). Carbonaceous aerosol species





on eighth section of each filter were oxidised quantitatively to $CO_2$ gas (Major et al., 2018).
This was later introduced into an IonPlus Enviro Mini Carbon Dating System spectrometer
(Switzerland) via its dedicated gas ion source interface. The measured results for the exposed
filters were corrected for the blank values. The $^{14}C/^{12}C$ ratios were also corrected for isotopic
fractionation by using the $^{13}C/^{12}C$ ratios (Wacker et al., 2010) that were obtained
simultaneously in the actual AMS measurements. The $^{14}C/^{12}C$ isotope ratios derived were also
normalised to that of the oxalic acid II 4990C standard reference material (NIST, USA), and
the measurement results were expressed as fraction of modern carbon ($f_m$), which denotes the
$^{14}C/^{12}C$ ratio of the samples relative to that of the unperturbed atmosphere in the reference year
of 1950 (Burr and Jull, 2009). Since majority of currently combusted fuel wood was growing
during the interval of atmospheric nuclear fusion bomb tests in the late 1950s and early 1960s,
the samples were corrected by a mean factor of 1.08 derived for the Northern Hemisphere
(Szidat et al., 2009; Heal et al., 2011). Thus, the fraction of contemporary carbon ($f_c$) was
calculated as $f_c = f_m/1.08$. The same correction factor was also adopted for the TC from biogenic
sources, although it is expected to show a somewhat smaller $^{14}C$ abundance. The differences
in the $f_c$ caused by the refined correction factor are ordinarily small when compared to the
method uncertainties (Minguillón et al., 2011) and, therefore, this effect was neglected.

A quarter section of each filter was utilized to determine the K (as a possible inorganic tracer
for BB), Ni (as a possible tracer for residual oil combustion) and Pb (as a former tracer for
vehicles with gasoline engine) content of the aerosol samples by inductively coupled plasma
optical emission spectrometry using an iCAP7400 DUO instrument (Thermo Fischer
Scientific, Germany). The filter sections were extracted by microwave-assisted $HNO_3$–$H_2O_2$
digestion. The analytical results for the exposed filters were corrected for the blank values. The
LOQ values of the elements listed were approximately 0.02 µg m$^{-3}$, 0.4 and 0.5 ng m$^{-3}$,
respectively, and most atmospheric concentration were above them.
**2.3 Data evaluation and modelling**
Concentrations of organic matter (OM) were derived from the OC data by a conversion factor
of 1.4 for the regional background and suburban area, and of 1.6 suggested for the city centres
(Turpin and Lim, 2001; Russell, 2003). It was estimated that the relative uncertainty associated
with the conversion is approximately 30% (Maenhaut et al., 2012). Whenever it was possible,
the comparisons of atmospheric concentration, other variables or their ratios with respect to
sites or seasons were accomplished by calculating first the ratios on a sample-by-sample or



day-by-day basis and then by averaging these individual ratios for the subset under
consideration.

The coupled radiocarbon-LVG marker method was utilised to apportion the TC among the
$EC_{FF}$, $OC_{FF}$, $EC_{BB}$, $OC_{BB}$ and $OC_{BIO}$ (Salma et al., 2017). The method consists of pragmatic
attribution steps, which are realised by multiplications with apportionment factors. The factors
are calculated for each sample from measured TC, $f_c$, EC, OC and LVG concentrations as
primary input data and from general, a priori known EC/OC ratio for BB [$(EC/OC)_{BB}$] and
OC/LVG ratio for BB [$(OC/LVG)_{BB}$]. They combined adaptation is related to subsequent and
step wise subtraction of contemporary TC, $EC_{BB}$ and $OC_{BB}$ from TC on the one hand, and of
$EC_{FF}$ from fossil TC on the other hand. The apportionment factors are expressed as: $f_1=f_c$,
$f_2=(OC/LVG)_{BB}\times LVG\times (EC/OC)_{BB}/f_1/TC$, $f_3=(OC/LVG)_{BB}\times LVG/f_1/(1–f_2)/TC$ and $f_4=(EC/TC$
$–f_1\times f_2)/(1–f_1)$ (Salma et al., 2017). For the $(EC/OC)_{BB}$ ratio, we implemented a mean of 17%
derived from a critically evaluated ratio and standard deviation (SD) of (16±5)% (Szidat et al.,
2006) and from a ratio and SD of (18±4)% (Bernardoni et al., 2011, 2013) obtained specifically
for wood burning. As far as the $(OC/LVG)_{BB}$ ratio is concerned, its actual value depends
predominantly on the wood types and burning conditions (Puxbaum et al., 2007). In the present
study, an $(OC/LVG)_{BB}$ ratio of 5.59 was adopted (Schmidl et al., 2008). The mean
apportionment factors separately for the different site types and seasons are summarised in
Table S1 in the Supplement. It is the $OC_{BIO}$ and $OC_{BB}$ which are most sensitively influenced
by the input uncertainties. Their relative uncertainty for some individual low concentrations
could be up to 40–50%, while it is expected to be approximately 30% or smaller for the other
carbonaceous species.
**3 Results and discussion**
General relationships that can exist among the atmospheric environments and seasons
including coupled meteorological and chemical processes need to be overviewed before
interpreting the spatial and temporal variability and tendencies in aerosol properties.
**3.1 Differences and similarities among atmospheric environments and seasons**
The median concentrations of $SO_2$, NO, $NO_2$ and $PM_{10}$ mass over the sampling time intervals
were larger in the city centre than in the suburban area in all seasons (Table S2 in the
Supplement). This can mainly be explained by their anthropogenic sources in the city centre,





mostly due to the increased intensity and density of road traffic. In contrast, the $O_3$ level was
substantially higher in the suburban area than in the city centre and considerably larger in the
regional background that in the suburban area. It tended to show a maximum in summer. This
behaviour is typical for large-scale $O_3$ formation mechanism. This all implies that there were
substantial differences in photochemical activity between the regional background and the
urban sites except for summer.

The meteorological data over the sampling time intervals are in accordance with ordinary
values and denote weather situations without extremes (Table S3 in the Supplement). The $T$
data indicate an urban heat island in central Budapest, particularly in winter and autumn. At
the regional site, there was snow cover with a thickness from 2 to 4 cm during the sample
collections in winter for approximately 4 days, while in the Budapest area, there was snow in
spots with a thickness of 1–2 cm for 2–3 days. The data suggest that there was somewhat milder
weather over the winter sample collections than usually present.

Time series of $PM_{2.5}$ mass, EC and WSOC over the sampling time intervals separately in the
different environments and seasons are shown in Figs. 2, 3 and 4, respectively. The former
variable represents the bulk fine PM; EC is a typical primary aerosol constituent, while WSOC
expresses the SOA. These species are rather different as far as their sources and properties are
concerned. Nevertheless, their concentrations often changed coherently at the locations with
the strongest link in winter. It can likely be explained by the common effects of regional
meteorology – particularly of boundary layer mixing height – in the Carpathian Basin
especially under anticyclonic weather situations. It seems that the daily evolution of the
regional meteorology and partially, the long-range transport of air masses often have higher
influence on the changes in atmospheric concentrations than the source intensities if the sources
are distributed over a large area (Salma et al., 2001, 2004). The strongest connection can be
related to cold air masses above the Carpathian Basin which generate a lasting $T$ inversion layer
(the so-called cold pillow) and which restricts the vertical mixing and results in poor air quality
over extended areas of the basin in larger and smaller cities as well as in rural areas.

**Figure 2.** Time variation of PM$_{2.5}$ mass for regional background in the Carpathian Basin, suburban area and city centre of Budapest during the aerosol sampling time intervals in different seasons (a–d).

**Figure 3.** Time variation of EC for regional background in the Carpathian Basin, suburban area and city centre of Budapest during the aerosol sampling time intervals in different seasons (a–d).

**Figure 4.** Time variation of WSOC for regional background in the Carpathian Basin, suburban area and city centre of Budapest during the aerosol sampling time intervals in different seasons (a–d).

## 3.2 Tendencies in aerosol concentrations

Median atmospheric concentrations of the measured aerosol constituents separately for the different environmental types and seasons are presented in Table 2. The concentrations are in line with or somewhat smaller than the corresponding results obtained in earlier studies at the same or similar locations usually for shorter time intervals (Salma and Maenhaut, 2006; Kiss et al., 2002; Salma et al., 2004, 2007, 2013, 2017; Ion et al., 2005; Maenhaut et al., 2005, 2008;





Puxbaum et al., 2007; Blumberger et al., 2019). The $PM_{2.5}$ mass and OC concentrations in the
city centre were larger by a mean factor of 1.6−1.7 than in the regional background, while they
were similar to the suburban data. Their winter values were usually the largest, and they
reached the minimum in summer or spring.

The concentrations of EC increased monotonically in the order of the environments: regional
background, suburban area and city centre location by typical factors of 2 and 3, respectively.
In the regional background, the EC data for autumn and winter were similar to each other and
they were the largest. In the suburban area, the EC data showed a maximum in winter and a
minimum in summer. In the city centre, the EC levels in autumn, winter and spring were similar
to each other, and they all showed a minimum in summer. These can be explained by larger
intensity of soot emissions from incomplete burning (road vehicles, residential heating and
cooking by solid fuel), which is a typical anthropogenic source, and which is associated with
seasonal variation (e.g. due to residential heating) as well as with constant sources (e.g. due to
traffic or cooking) over a year.

The WSOC showed maximum medians in winter at all sites. In autumn and summer, the urban
locations had similar concentrations to each other, while it was somewhat smaller in the
regional background. In winter, the suburban site exhibited the maximum median
concentration. This is explained by larger influence of BB in this environment and season and
by higher water solubility of its products (see later and Sects. 3.3 and 3.5). In spring, the
medians had a monotonically increasing tendency from the regional background to the city
centre.





**Table 2.** Median atmospheric concentrations of PM$_{2.5}$ mass, elemental carbon (EC), organic carbon
(OC), water-soluble organic carbon (WSOC), levoglucosan (LVG), mannosan (MAN), galactosan
(GAN), fraction of contemporary total carbon ($f_c$), K, Ni, Pb and total aerosol particle number ($N_{6-1000}$)
for regional background in the Carpathian Basin, suburban area and city centre of Budapest in different
seasons.

| Constituent | Site type | Autumn | Winter | Spring | Summer |
|---|---|---|---|---|---|
| PM$_{2.5}$ mass | Region | 12.5 | 16.5 | 8.6 | 10.7 |
| ($\mu$g m$^{-3}$) | Suburb | 25 | 26 | 9.7 | 11.7 |
| | Centre | 28 | 24 | 13.3 | 8.2 |
| EC | Region | 0.37 | 0.36 | 0.20 | 0.122 |
| ($\mu$g m$^{-3}$) | Suburb | 0.45 | 0.68 | 0.51 | 0.35 |
| | Centre | 0.99 | 0.77 | 0.79 | 0.37 |
| OC | Region | 2.3 | 3.2 | 2.0 | 2.2 |
| ($\mu$g m$^{-3}$) | Suburb | 4.5 | 5.4 | 2.4 | 2.7 |
| | Centre | 6.6 | 4.6 | 2.8 | 2.6 |
| WSOC | Region | 1.63 | 2.0 | 1.08 | 1.66 |
| ($\mu$g m$^{-3}$) | Suburb | 2.4 | 3.8 | 1.27 | 2.0 |
| | Centre | 2.7 | 2.8 | 1.45 | 2.0 |
| LVG | Region | 0.172 | 0.40 | 0.0180 | 0.0081 |
| ($\mu$g m$^{-3}$) | Suburb | 0.44 | 0.71 | 0.040 | 0.0124 |
| | Centre | 0.38 | 0.48 | 0.036 | 0.0103 |
| MAN | Region | 19.2 | 18.5 | 2.6 | <1.2 |
| (ng m$^{-3}$) | Suburb | 37 | 39 | 2.9 | <1.2 |
| | Centre | 26 | 21 | 4.1 | <1.2 |
| GAN | Region | n.a. | 10.6 | 1.20 | <0.5 |
| (ng m$^{-3}$) | Suburb | 16.4 | 20 | 0.61 | <0.5 |
| | Centre | 11.7 | 14.1 | 1.21 | <0.5 |
| $f_c$ | Region | 69 | 75 | 61 | 74 |
| (%) | Suburb | 66 | 74 | 48 | 60 |
| | Centre | 76 | 74 | 48 | 60 |
| K | Region | 0.182 | 0.23 | 0.088 | 0.081 |
| ($\mu$g m$^{-3}$) | Suburb | 0.22 | 0.25 | 0.097 | 0.075 |
| | Centre | 0.26 | 0.27 | 0.106 | 0.057 |
| Ni | Region | 0.75 | 0.68 | 1.21 | 1.12 |
| (ng m$^{-3}$) | Suburb | 0.88 | 0.78 | 1.24 | 1.09 |
| | Centre | 1.10 | 0.63 | 1.51 | 1.08 |
| Pb | Region | 3.8 | 3.0 | 3.2 | 2.8 |
| (ng m$^{-3}$) | Suburb | 5.8 | 6.8 | 4.1 | 3.5 |
| | Centre | 7.5 | 5.2 | 4.4 | 2.3 |
| $N_{6-1000}$ | Region | 3.9$^{\dagger}$ | 3.7$^{\dagger}$ | 4.8$^{\dagger}$ | n.a. |
| (10$^3$ cm$^{-3}$) | Suburb | n.a. | n.a. | n.a. | n.a. |
| | Centre | 14.6 | 10.9 | 13.9 | 6.4 |

n.a.: not available; †: for $N_{6-800}$





The mean atmospheric concentrations of the monosaccharide anhydrides were decreasing in
the order of LVG, MAN and GAN. The concentrations of LVG were larger by ca. 1 order of
magnitude than for the joint concentrations of MAN and GAN. Their mean ratio was the largest
in winter and the smallest in autumn. This could be affected by the share of hardwood burnt in
different seasons (Fine et al., 2004; Schmidl et al., 2008; Maenhaut et al., 2012). The LVG
concentration did not vary monotonically with respect to the sites; it was larger in the city
centre by a factor of 1.7 than in the regional background and was smaller by approximately
20% than in the suburban area. This could be related to the spatial distribution of biofuel
utilisation mainly for residential heating and to atmospheric dispersion of their emission
products in the different environments.

As far as the contemporary C is concerned, there were three individual consecutive samples
collected in the city centre in autumn with significantly larger values than any other data in the
set. There are several applications of nuclide $^{14}$C mostly in pharmaceutical/medical and
biological academy field in Budapest, which could release radiocarbon of anthropogenic origin
into the ambient air (in particular from labelled inorganic compounds such as $NaHCO_3$). These
three data were regarded to be outliers and were excluded from the further evaluation. The
centre/suburb $f_c$ ratio in autumn, however, remained still somewhat higher (1.15) with respect
to the other seasons (for which the ratios were uniformly 1.00). This indicates that the
anthropogenic $^{14}$C contamination could lightly affect the remaining analytical results as well.
Its consequences on the source apportionment are discussed in Sect. 3.5. In autumn and winter,
the mean centre/suburb, centre/region and suburb/region ratios were similar to each other (with
an overall mean and SD of 1.02±0.10) at all sites, while in spring and summer; they decreased
in the order of the ratios above with means and SDs of 1.04±0.20, 0.82±0.13 and 0.80±0.15,
respectively. These tendencies are governed by carbonaceous matter of different origin.

The concentrations of K in autumn, winter and spring increased monotonically for the regional
background, suburb area and city centre. They showed a maximum in winter. Its concentrations
were the smallest in summer and exhibited an opposite tendency as far as the location types are
concerned, thus they decreased monotonically for the sites listed above. The concentrations of
Ni were similar to each other without any evident tendency. Except for its concentrations in
spring, which seemed to be the largest. The concentration of Pb showed an increasing tendency
from the regional background to the urban sites. The present data are smaller than the median
levels of 16 ng m$^{-3}$ in the city centre and of 9 ng m$^{-3}$ in the near-city background measured in





spring 2002 after the phase out of leaded gasoline in Hungary in April 1999, and are in line
with its overall decreasing trend (Salma and Maenhaut, 2006; Salma et al., 2000).

The particle number concentrations in the city centre were 1 order of magnitude larger than in
the regional background. This is explained by the spatial distribution of their main sources and
by the relatively short atmospheric residence time of ultrafine particles (Salma et al., 2011).
The latter property also causes that there are larger variations in their atmospheric
concentrations, which implies that the present measurement (sampling) time intervals are not
long enough for reliable conclusions on tendencies.

### 3.3 Tendencies in concentration ratios

Mean values and SDs of some important concentration ratios separately for the different
environments and seasons are shown in Table 3. The $PM_{2.5}/PM_{10}$ mass ratio exhibited strong
seasonal dependency. In spring and summer, the $PM_{10}$–$PM_{2.5}$ fraction particles (coarse mode)
made up approximately 2/3 of the particulate mass, while in autumn and perhaps also in winter,
the $PM_{2.5}$ mass prevailed with a similar ratio. These imply and confirm that in spring and
summer, the suspension or resuspension of soil, crustal rock, mineral and roadside dust is
substantial in Budapest, while in autumn and winter, the aerosol mass levels are more
influenced by residential heating, cooking and road traffic (Salma and Maenhaut, 2006).

Contribution of the OM to the $PM_{2.5}$ mass for the regional background, suburban area and city
centre showed little seasonal variation with annual means and SDs of $(31\pm5)\%$, $(32\pm6)\%$ and
$(35\pm7)\%$, respectively. These balanced contributions are in line with other European results
(Puxbaum et al., 2007; Putaud et al., 2010), and indicate a huge number, big variety and
spatially more-or-less equally distributed sources of OC in the Carpathian Basin. The mean
contributions of EC to the $PM_{2.5}$ mass were between 1 and 6%, with a minimum in the regional
background in summer. The contributions can change substantially in different
microenvironments within a city (e.g. 14% for a street canyon in central Budapest in spring;
Salma et al., 2004; Maenhaut et al., 2005). The carbonaceous particles (OM+EC) in the
regional background, suburban area and city centre made up $(32\pm5)\%$, $(36\pm7)\%$ and $(39\pm7)\%$,
respectively of the $PM_{2.5}$ mass as annual means and SDs. Their seasonal means revealed limited
variability (except for the city centre, where it changed from 33% in winter to 48% in summer).
The $TC/PM_{2.5}$ mass ratios are given as auxiliary information to allow the recalculation of the
contributions to the TC shown in Fig. 5 to that to the $PM_{2.5}$ mass.






**Table 3.** Mean values and SDs for the $PM_{2.5}/PM_{10}$ mass, $OM/PM_{2.5}$ mass, $EC/PM_{2.5}$ mass, $TC/PM_{2.5}$ mass, WSOC/OC and OC/EC ratios for regional background in the Carpathian Basin, suburban area and city centre of Budapest in different seasons.


| Ratio | Site type | Autumn | Winter | Spring | Summer |
|---|---|---|---|---|---|
| $PM_{2.5}/PM_{10}$ | Region | 64±10 | n.a. | n.a. | n.a. |
| mass | Suburb | 64±4 | 67±11 | 30±8 | 48±8 |
| (%) | Centre | 67±9 | 56±14 | 32±4 | 33±7 |
| $OM/PM_{2.5}$ | Region | 33±6 | 29±5 | 32±5 | 33±4 |
| mass | Suburb | 32±9 | 31±5 | 32±6 | 32±5 |
| (%) | Centre | 36±5 | 30±3 | 30±4 | 36±5 |
| $EC/PM_{2.5}$ | Region | 3.1±1.4 | 2.3±0.5 | 2.3±0.7 | 1.2±0.3 |
| mass | Suburb | 3.2±1.0 | 3.1±0.9 | 4.9±2.2 | 3.5±1.1 |
| (%) | Centre | 4.3±2.4 | 3.3±0.6 | 6.4±1.7 | 4.6±0.9 |
| $TC/PM_{2.5}$ | Region | 28±5 | 22±4 | 25±4 | 23±3 |
| mass | Suburb | 24±7 | 26±4 | 27±5 | 26±4 |
| (%) | Centre | 27±5 | 22±3 | 28±5 | 29±10 |
| WSOC/OC | Region | 64±11 | 58±7 | 54±9 | 72±5 |
| (%) | Suburb | 55±16 | 64±6 | 53±7 | 76±6 |
| | Centre | 42±16 | 59±4 | 56±9 | 76±11 |
| OC/EC | Region | 8.8±3.2 | 9.0±1.4 | 11±3 | 18±4 |
| | Suburb | 8.4±3.5 | 7.3±1.5 | 5.4±2.0 | 7.3±1.6 |
| | Centre | 6.3±2.2 | 5.9±0.7 | 3.5±0.7 | 6.7±1.5 |


The mean WSOC/OC ratios in autumn showed a monotonically increasing tendency from the city centre to the regional background. This is just opposite to the atmospheric WSOC concentration (which decreased monotonically). In winter, the suburban area exhibited the maximum concentration. This can be explained by intensive BB in the area with respect to the other environments (Sect. 3.5) and with the fact that BB particles possess relatively high hygroscopicity (Swietlicki et al., 2008) and water solubility. In the remaining two seasons, the shares of the WSOC were similar to each other and varied without an obvious tendency. This can be linked to comparable and large photochemical activity in all environments in spring and summer (Sect. 3.1). The present ratios are in line with the values reported earlier for the corresponding locations (Kiss et al., 2002; Ion et al., 2005; Maenhaut et al., 2005, 2008; Viana et al., 2006; Puxbaum et al., 2007; Salma et al., 2007). It is noted that the determined OC (and WSOC) concentrations are somewhat method dependent; their ratios can change sensitively e.g. with the thermal protocol used in the OC/EC TOT analyser for samples containing large amounts of refractory C (Kuhlbusch et al., 2009; Pantheliadis et al., 2015).




The highest OC/EC ratios indicate the conditions under which the SOA formation is the largest.
The ratio had a maximum in the regional background in summer, which can be associated with
large photochemical activity and strong GRad. The ratios for the urban locations did not
indicate obvious seasonal tendencies. Formation, composition and properties of SOA and
atmospheric humic-like substances (HULIS) together with modelling the air mass transport
within the Carpathian Basin are to be dealt with in a separate paper after additional
investigations are completed.

Finally, it is noted for completeness that the annual mean LVG/MAN ratios and SDs for the
regional background, suburban area and city centre were $13.9\pm5.9$, $14.3\pm6.2$ and $14.7\pm5.8$,
respectively, and that ca. 40% of all available individual ratios were larger than the limit of
14.8 derived by Schmidl et al. (2008). The latter value was obtained for the combustion of
common hardwood (beech and oak) and softwood species (spruce and larch) in domestic wood
stoves in Austria. This means for our samples and conditions, the relationship between the
softwood and hardwood burnt mentioned is not applicable because of several reasons, e.g. the
likely differences in fireplaces and fuel wood in Hungary and mid-European Alpine regions.
**3.4 Apportioned carbonaceous species**
Median atmospheric concentrations of the apportioned $EC_{FF}$, $EC_{BB}$, $OC_{FF}$, $OC_{BB}$ and $OC_{BIO}$
aerosol constituents derived by the coupled radiocarbon-LVG model separately for the
different environments and seasons are summarised in Table 4. The present values are coherent
with the earlier median concentration from late winter/early spring of 2014 at the BpART
Laboratory (Salma et al., 2017) and with the results for the regional background (Gelencsér et
al., 2007; Puxbaum et al., 2007). The uncertainty of the individual apportioned data could be
larger than for the experimental results (e.g. TC) and, therefore, the substantial differences
among their means and their obvious tendencies are only interpreted.

The median concentrations of $EC_{FF}$ were similar to each other in autumn, spring and summer,
and exhibited a minimum in winter. In all seasons, its concentrations in the urban environments
tended to be larger by a factor of 2–3 than in the regional background. The $OC_{FF}$ concentrations
at the urban locations were similar to each other in all seasons, while they tended to be larger
than the regional values by a factor of 2–3 in autumn and summer. The $EC_{BB}$ and $OC_{BB}$
concentrations showed a maximum in winter and a minimum in summer. The concentrations



of $OC_{BB}$ in the city centre seemed to be somewhat smaller than in the suburban area, while the
latter was larger by a factor of 2–3 than in the regional background in autumn and spring. The
concentrations of $OC_{BIO}$ showed a monotonically decreasing tendency from autumn to
summer, spring and winter in all environments. The fluxes of BVOCs from plants strongly
depend on environmental conditions, age of leaves and vegetation, water and nutrient
availability, and it is also affected by the presence of some anthropogenic emissions.
Photochemical oxidation reactions of BVOCs, interactions among biogenic and anthropogenic
precursors and products, and aerosol formation yield considerations play a rather important
role in the process (McFiggans et al., 2019). The tendencies are further discussed after deriving
the contributions of the apportioned species to various quantities in Sect. 3.5.

**Table 4.** Median atmospheric concentration of apportioned elemental carbon from fossil fuel
combustion ($EC_{FF}$) and from biomass burning ($EC_{BB}$), of apportioned organic carbon from fossil fuel
combustion ($OC_{FF}$), from biomass burning ($OC_{BB}$) and from biogenic sources ($OC_{BIO}$) in µg m$^{-3}$ for
regional background in the Carpathian Basin, suburban area and city centre of Budapest in different
seasons.

| Constituent | Site type | Autumn | Winter | Spring | Summer |
|---|---|---|---|---|---|
| $EC_{FF}$ | Region | 0.35 | 0.057 | 0.23 | 0.12 |
| | Suburb | 0.35 | 0.10 | 0.57 | 0.32 |
| | Centre | 0.60 | 0.24 | 0.74 | 0.36 |
| $EC_{BB}$ | Region | 0.19 | 0.34 | 0.020 | 0.0076 |
| | Suburb | 0.40 | 0.62 | 0.050 | 0.0083 |
| | Centre | 0.36 | 0.46 | 0.047 | 0.0095 |
| $OC_{FF}$ | Region | 0.85 | 1.0 | 0.71 | 0.53 |
| | Suburb | 2.1 | 1.1 | 1.0 | 0.83 |
| | Centre | 1.5 | 1.2 | 1.0 | 0.81 |
| $OC_{BB}$ | Region | 1.1 | 2.0 | 0.12 | 0.045 |
| | Suburb | 2.4 | 3.6 | 0.29 | 0.049 |
| | Centre | 2.1 | 2.7 | 0.27 | 0.056 |
| $OC_{BIO}$ | Region | 2.0 | 0.22 | 1.3 | 1.8 |
| | Suburb | 2.3 | 0.36 | 1.2 | 1.8 |
| | Centre | 3.1 | 0.31 | 1.3 | 1.6 |


Pearson's coefficients of correlation between the variables were calculated to examine their
possible paired relationships. The results should be interpreted with caution since many data
sets are not (fully) independent from each other and can be biased by meteorological processes
(Sect. 3.1), can be coupled by their potential common sources or can be influenced jointly by





further factors/causes for them. Moreover, interactions among biogenic and anthropogenic
VOCs or among organic precursors with rather different SOA yields can significantly enhance
or suppress, respectively the SOA production (Hoyle et al., 2011; McFiggans et al., 2019).
Selected coefficients are shown in Table S4 in the Supplement for the annual data sets.
Potassium correlated with both carbonaceous species of BB origin at all locations, while its
coefficients with the other variables seemed insignificant. There was a linear relationship
between NO (which is emitted for 60–70% by road vehicles in Budapest) and $OC_{FF}$. The
relationships between $T$ and the apportioned constituents indicated that BB was more intensive
under cold weather conditions, while the utilisation of FFs was more constant over the year
(campaign). The relationships approved the correct attribution of the apportioned species.

### 3.5 Contributions of major source types

It was the FF combustion that showed the most balanced and constant daily or seasonal mean
contributions to the TC over the whole year and at all sites. Its annual means and SDs for the
regional background, suburban area and city centre were (31±7)%, (36±12)% and (36±13)%,
respectively. In contrast, the daily mean contributions of BB and biogenic sources changed
radically over the seasons at all locations. For BB, the individual contributions for the
atmospheric environments listed above ranged from <2 to 73% (with a median of 10%), from
<2 to 73% (24%) and from <2 to 72% (19%), respectively. The analogous daily data for
biogenic sources spanned from <2 up to 88% (52%), from <2 to 70% (35%) and from <2 to
67% (39%), respectively.

Their seasonal mean contributions in forms of EC and OC to the TC separately for the different
environmental types are shown as circle chart diagrams in Figure 5. In autumn, the three major
source types contributed equally to the TC. In winter, it was the BB which was the major source
with a relative share of approximately 60% at all sites, and its contribution was the largest in
this season. The contributions of FF combustion in winter were similar to each other for all site
types with a typical share around 25%. The contributions of biogenic sources were the smallest
in this season, although they were still non-negligible. Their share showed an increasing
tendency from the regional background to the urban sites (from 5 to 8%), which could likely
be explained by larger temperatures (urban heat island) and less snow coverage in the city
centre (Sect. 3.1) than in its surroundings. In spring, FF combustion and biogenic sources were
the largest two contributors at all locations with typical shares of 45–50% each. The $EC_{FF}$
showed the largest contributions in spring, which were increased monotonically in the order of



the location type: region, suburb and centre. In summer, biogenic sources became the major
contributor with a monotonically increasing share from the centre to the region.

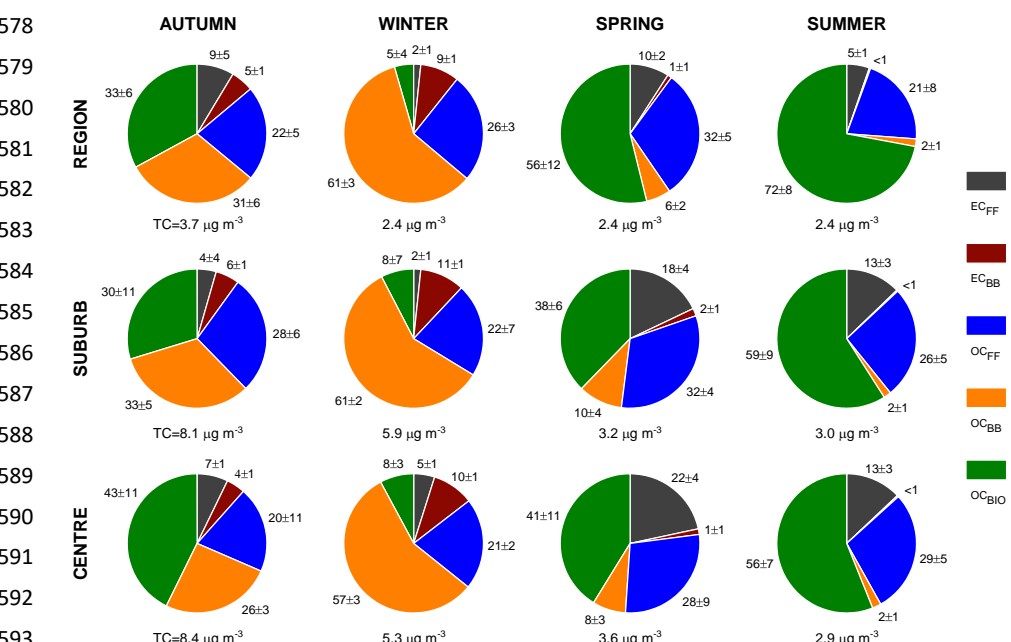

**Figure 5.** Mean contributions with SDs of elemental carbon from fossil fuel combustion (EC$_{FF}$) and
from biomass burning (EC$_{BB}$), of organic carbon from fossil fuel combustion (OC$_{FF}$), from biomass
burning (OC$_{BB}$) and from biogenic sources (OC$_{BIO}$) to PM$_{2.5}$-fraction total carbon (TC) in % for regional
background in the Carpathian Basin, suburban area and city centre of Budapest and seasons. The median
atmospheric concentrations of TC are indicated under individual circle charts, while the corresponding
mean TC/PM$_{2.5}$ mass ratios are shown in Table 3.

Further conclusions can be derived by focusing on specific contributions of EC$_{FF}$ and EC$_{BB}$ to
EC, and of OC$_{FF}$, OC$_{BB}$ and OC$_{BIO}$ to OC (Figs. 6 and 7, respectively). Elemental carbon is
sometimes applied as a marker of automotive emissions mainly from diesel engines in cities of
the continental mid-latitude northern hemisphere. The present research indicates that in urban
ambient air in Central Europe, this assumption is, however, valid only in spring and summer
(when the share of the EC$_{FF}$ was indeed larger than 90%). In autumn, the contributions of EC$_{BB}$
can be considerable (up to 40–60%) at urban sites, so they can be by no means negligible.
Furthermore, in winter, the relative mass of soot particles from BB can be even larger.



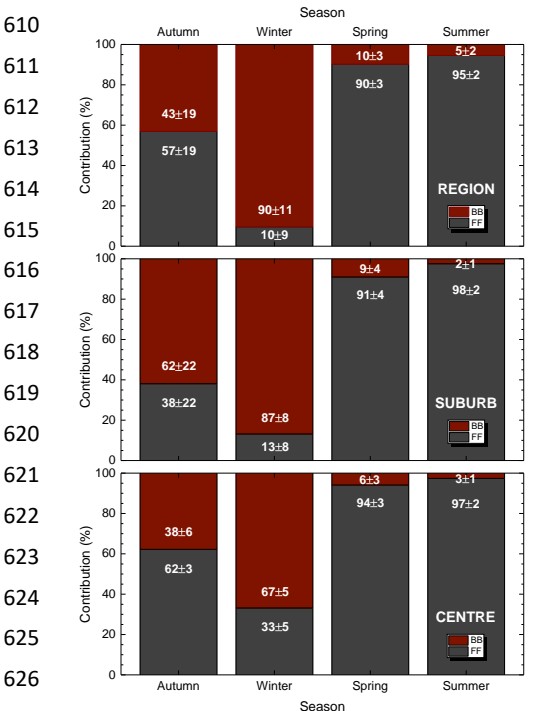

**Figure 6.** Seasonal distribution of mean contributions from FF combustion and BB with SDs to the PM$_{2.5}$-fraction EC for regional background in the Carpathian Basin, suburban area and city centre of Budapest. The corresponding median atmospheric concentrations of the EC are given in Table 2.

**Figure 7.** Seasonal distribution of mean contributions from FF combustion, BB and biogenic sources (BIO) with SDs to the PM$_{2.5}$-fraction OC for regional background in the Carpathian Basin, suburban area and city centre of Budapest. The corresponding median atmospheric concentrations of OC are given in Table 2.

It can be seen in Fig. 7 that the contributions from FF combustion to OC were fairly constant with a typical value of 25–35% and with an overall mean and SD of (30±8)% averaged for all atmospheric environments and seasons. In spring, some possible elevation could occur. Biomass burning was the major source of OC with a share of 67% in winter. Biogenic sources prevailed (with a share of 65–75%) in summer and made up about half of the OC in spring. The contributions from BB were hardly quantifiable in summer, while biogenic emissions were still considerable in winter, particularly at the urban sites. In autumn, the three major source types made up balanced (constant and similar) contributions in all environments. Closely looking, there could be a slight overestimation of biogenic sources and a related small





underestimation of FF combustion (see Fig. 3 in Salma et al., 2017) in the city centre (left lower
column in Fig. 7) due to possible atmospheric contamination by anthropogenic $^{14}$C, which was
discussed in Sect. 3.2. It is mentioned that primary organic aerosol is not included in the model,
which could influence somewhat the overall contributions of the sources.
**3.6 Potentials for air quality**
To examine the potentials of the apportioned carbonaceous species for regulatory and
legislation purposes, the contributions of the main source types to the PM$_{2.5}$ mass were roughly
estimated. It was assumed that the OM/OC conversion factors for the aerosol particles
originating from FF combustion, BB and biogenic sources were equal to the conversion factor
for the bulk fine-fraction particles, thus 1.4 for the regional background and suburban area, and
1.6 for the city centre. The results obtained are summarised in Table S5 in the Supplement. The
separate contributions typically represent up to 1/5 or 1/4 of the PM mass as lower estimates
and are discussed in Sect. 4.

The contributions were also evaluated as function of the PM$_{2.5}$ mass concentration, which is a
key measure for air quality considerations. The plots for the seasonal mean contributions of
OC$_{FF}$, OC$_{BB}$ and OC$_{BIO}$ to the TC are shown in Fig. 8a–c, respectively. The contributions of
FF combustion (Fig. 8a) did not seem to depend substantially on the PM$_{2.5}$ mass level at any
of the locations, so FF exhibits a constant and steady-state importance over various air pollution
periods. The share of BB showed an increasing tendency with poor air quality (Fig. 8b). The
change rate (the slope of the fitted line, $b$) was larger for the regional background ($b$=6.7) and
smaller but similar to each other for the urban sites ($b\approx2.1$). For biogenic emissions the trends
were just the opposite (Fig. 8c); their relative importance decreased by poorer air quality. The
tendency was similar again for the urban sites ($b\approx-1.5$) and substantially larger for the regional
background ($b$=–6.9). The tendencies of the EC$_{FF}$ and EC$_{BB}$ were analogous. These results
together indicate that BB influences the air quality in the regional background very extensively
and it also has substantial effect on the air quality in the Budapest area, mainly in winter and
in autumn as well. The conclusions have importance in and consequences on the potentials for
improving the air quality further interpreted in Sect. 4.



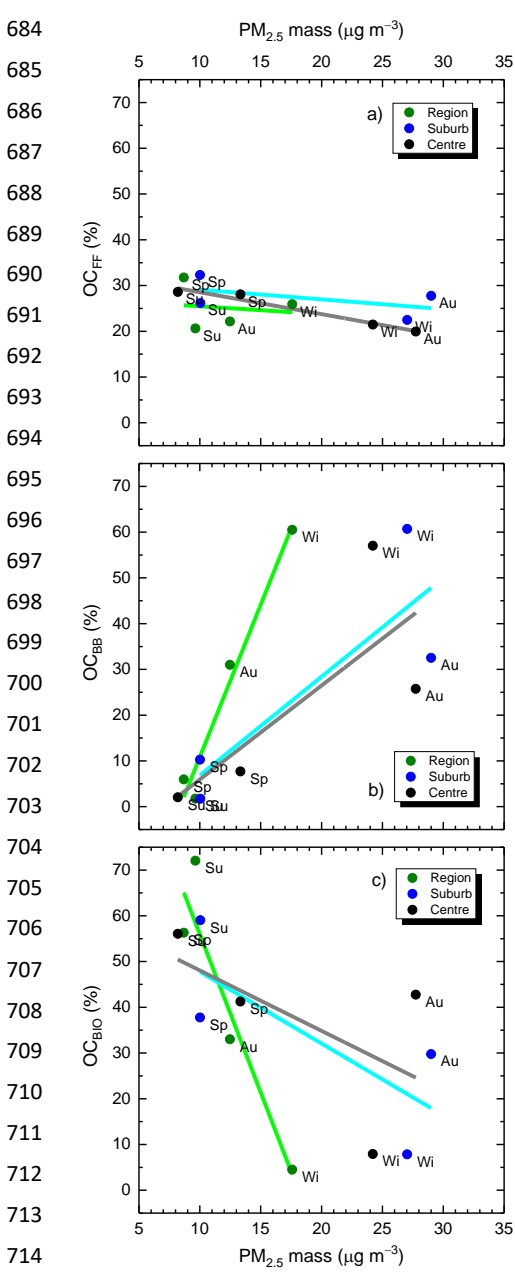

**Figure 8.** Seasonal mean contribution of apportioned $OC_{FF}$ (a), $OC_{BB}$ (b) and $OC_{BIO}$ (c) to TC as function of the seasonal median $PM_{2.5}$ mass concentration (surrogate or proximity value for the air quality) for regional background in the Carpathian Basin, suburban area and city centre of Budapest. The seasons are indicated by their two starting letters. The linear lines are to guide the eyes.





### 4 Conclusions

In the present study, the major carbonaceous aerosol species were apportioned among FF combustion, BB and biogenic sources in various types of atmospheric environments of interest in the Carpathian Basin in each season. The results and conclusions achieved were obtained from the first systematic complex research project as far as the spatial scale within the basin and time span (of 1 year) are concerned and represent valuable research contributions on a large area in Central Europe.

The carbonaceous particles made up from 30 to 48% of the $PM_{2.5}$ mass (as seasonal mean) depending on the environment and season. It is the BB in winter that represents the largest potential (with a mass share of >20%; Table S4 in the Supplement) for improving the air quality both in cities and on rural areas of the basin. It is worth mentioning that all air pollution (smog) alert episodes in Hungary were announced so far exclusively because of the $PM_{10}$ mass limit exceedances and they all happened in winter. Possibilities in controlling various forms of BB for air quality improvements seem to be, therefore, rather relevant. In the present case for instance, there were 3, 8 and 8 days, respectively (19 days in total) in the subset of $4 \times 7$ days in the regional background, suburban area and city centre which daily mean values exceeded the EU annual $PM_{2.5}$ limit value of 25 µg m$^{-3}$. They all occurred in winter and autumn. If the BB sources (i.e. $OC_{BB}$ and $EC_{BB}$) had decreased by half of their actual concentrations then the number of exceedance days would reduce to 2, 6 and 5, respectively (13 days in total), while a perfect fuel gas aftertreatment of the BB as a sources would result in the number of exceedances of 1, 4 and 5, respectively (10 days in total). In addition to carbonaceous particles, some adjunct inorganic constituents are also generated and, more importantly, soil or mineral dust and fly ash particles are also mobilised or blown up into the air due to the combustion or burning process itselfs. These, on the one hand, can further and substantially enhance the overall mass contributions and potentials of the high-temperature sources (including BB), and, on the other hand, may change somewhat their relative contributions.

Fossil fuel combustion is an abundant source of PM mass (with a share of >20%; Table S4 in the Supplement) only at urban sites in spring and summer. Resuspension or suspension of road and surface dust by moving vehicles can again represent a substantial auxiliary increment for FF contribution. Biogenic sources are normally considered as natural process or to be dominated by natural processes, and, therefore and strictly speaking, they are not associated



with the issue of air pollution. It is expected that the unaccounted $PM_{2.5}$ mass contains
secondary inorganic aerosol particles mostly sulfates, nitrates and elements, and soil or
mineral/crustal rock dust particles as well (Salma et al., 2001), which should definitely be
revisited and taken into account in further source apportionment research.

Another challenge in health-related or air-quality-type assessment studies is to refine the
apportionment within the major source types with burning of plastics, domestic waste (garbage)
and household coal and fuel wood burning through identification of their appropriate tracers
and via quantification of various emission factors of their specific sources e.g. by advanced
hyphenated MS or optical methods combined with powerful statistical data treatment. These
additional combustion categories deserve more investigations since many of them seem to be
prevalent and of increasing volume in the studied geographical area, and they produce some
specific air pollutants or toxics which can represent serious risk for human health, wellbeing
and the environment.
**Data availability.** Raw data are available from the corresponding author on reasonable request.
**Supplement.** The supplement related to this article is available online.
**Competing interests.** The authors declare that they have no conflict of interest.
**Acknowledgements.** The authors are grateful to Gergő Farkas, Veronika Varga, Debóra Varga and
Péter Varga of the Eötvös University for their help in collecting the aerosol samples and their chemical
analyses. The map in Fig. 1 was created by Mátyás Gede of the Eötvös University, Department of
Cartography and Geoinformatics. Funding by the National Research, Development and Innovation
Office, Hungary (K116788 and K132254) is acknowledged. The research was supported by the
European Union and the State of Hungary, co-financed by the European Regional Development Fund
in the projects of GINOP-2.3.2-15-2016-00028 and GINOP-2.3.2-15-2016-00009 'ICER'. The project
was realised within the frame of the COST COLOSSAL Action (CA16109).

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
