# Peer review of "Fossil fuel combustion, biomass burning and biogenic sources"

_Atmospheric Chemistry and Physics, 2019_

## Referee Comment (RC1) · Anonymous Referee #1 · 19 Dec 2019

The authors have studied the influence of different anthropogenic and biogenic particle sources and meteorology in Carpathian basin for a year. The samples were analyzed for PM2.5 mass, organic and elemental carbon, water-soluble OC, radiocarbon, lev-oglucosan and some elements. Radiocarbon-LVG marker method was applied to apportion the total carbon (TC=OC+EC) into contributions of EC and OC from fossil fuel combustion, from biomass burning and from biogenic sources. Topic is interesting and source apportionment based on in-depth chemical analysis and radiocarbon method has scientific novelty value.

General comments:

-English language should be thoroughly checked by a native speaker. Especially in the introduction some sentences are a bit hard to understand.

-Chapter 2.1. It would help reader if you could name the stations in this chapter and clearly state stations called hereafter xx, zz and yy. Now there is many kind of variations of the names in the text. Maybe also include a more in-depth description of the area where stations are located (what kind of area, how many inhabitants, how much traffic/biomass burning/industry etc is in area, any prior knowledge about the expected sources?). Maybe the distance between stations or a map would help reader also. Please, add for all measured parameters the instrument, model and manufacturer. Also, it is bit hard to understand where e.g. DMPS was measuring and how long.

-Check and explain the used acronyms and terms. E.g. for elemental carbon both EC and soot are used, which can be very confusing to some of the readers. Also, terms carbonaceous aerosol, total carbon should be explained.

-More literature references should be added to the text to further discuss the results and their significance. Also more discussion about where these results could be utilized would be useful. The novelty value of results should be highlighted more!

Tables 2,3: The sampling periods for all stations are different. At the Central station the sampling period is much shorter. Are the mean/meadian values and ratios calculated for all the samples or only for the seven simultaneously collected samples? if sampling times are not exactly same, is it fair to compare the results of stations? e.g. some episode could change the concentrations and affect the observed mean values significantly.. if this episode is only included in longer timeseries measured in Background/suburban areas, this could affect the comparison when the results of different stations are compared.

Minor comments

Line 35: define carbonaceous

Line 39; define soot (as there is also EC)

Line s43-51: Sentences are bit long and hard to read. Clarify this and maybe specify if these consequences in the list are positive or negative in the nature.

Line 52: "Fuel wood"? does this refer to biomass combustion in residential scale?

Line 59-61: "Huge number, composite character, spatial and temporal variability of the sources together with the complex mixture and atmospheric transformation of their products make the quantification of these source types or their inventory-based source assessment challenging" Clarify this sentence, it is bit hard to understand.

Line 62-63: "There are several methods to apportion the particulate matter (PM) mass or carbonaceous species among some or all major source types." please clarify this sentence

Lines 63-69. Sentence is really long, maybe split to 2-3 shorter sentences?

Line 74: "The latter molecule is often applied together with its stereoisomers mannosan (MAN) and galactosan (GAN) since.." maybe change to "Monosaccharide anhydride analysis often contains stereoisomers mannosan (MAN) and galactosan (GAN) in addition to levoglucosan since..

Line 100: what is the "latter type" referring to? please clarify

Table 1. Why there is extra space between date and month as well as between the month and year in all timeperiods? please check the journal instructions how to give the dates..

Lines 152-167: Please add the model and manufacturer for all the instruments, provide the instrument information for the meteorological data as well information in which stations these instruments were used. E.g. was DMPS run in all stations constantly, or was one dmps moved between the stations? Maybe a table with station, instruments,

СЗ

models and measured components would help reader to understand the situation.

Line 169: Add balance model and manufacturer

Line 173: where does this LOQ value for PM mass comes from?

Line 198: what is origin of the LVG observed in the blank filters? please add how much levoglucosan was observed in the blanks.. has this kind of blank values seen in other studies?

Line 203: which days?

Line 236-240: "Whenever it was possible, the comparisons of atmospheric concentration, other variables or their ratios with respect to sites or seasons were accomplished by calculating first the ratios on a sample-by-sample or day-by-day basis and then by averaging these individual ratios for the subset under consideration". Please explain what variables/ratios this refers to?

Line 275: Maybe add some values for average temperature, wind etc meteorological parameters to article also (not only supplement) as people not living in Budabest may not know the normal local conditions mean.

Line 284-286: "The former variable represents the bulk fine PM; EC is a typical primary aerosol constituent, while WSOC is expresses the SOA." Sentence is hard to understand, please clarify what this means.

Chapters 3.2-3.6 please add some numberical values to text also. Would be also useful to compare more to literature wether the values were as expected or maybe lower/higher..

Line 391-395: maybe this information should be in experimental chapter?

Line 359-365; "These can be explained by larger intensity of soot emissions from incomplete burning (road vehicles, residential heating and cooking by solid fuel), which is a typical anthropogenic source, and which is associated with seasonal variation (e.g. due to residential heating) as well as with constant sources (e.g. due to traffic or cooking) over a year." what is the seasonal variation observed for road vehicles? how they differ from traffic that does not have seasonal variation?

Line 501-502: Did temperature have similar trend as OCbio? I would have expected to see highest OCbio concentrations in summer.. or is there a reason why autumn OCbio was higher? how does this observation compare to other studies?

Chapter 3.6 and conclusions: the impact of results is now discussed from the air quality point of view. Maybe add something about climate and health point of view also? assumably the anthropogenic emissions and BC have different, likely more negative health impacts. BC has also strong climate impact.

---

## Referee Comment (RC2) · Anonymous Referee #4 · 21 Feb 2020

Interactive comment on a paper by: Imre Salma, Anikó Vasanits-Zsigrai, Attila Machon, Tamás Varga, István Major, Virág Gergely, and Mihály Molnár: Fossil fuel combustion, biomass burning and biogenic sources of fine carbonaceous aerosol in the Carpathian Basin

General comments: The paper deals with source attribution of carbonaceous aerosols in the Carpathian basin using chemical analysis of PM2.5 samples obtained using high-volume samplers. Besides standard analytes as organic carbon (OC), elemental carbon (EC), water-soluble carbon (WSOC), and few chemical elements, wood combustion markers as levoglucosan (LVG), mannosan, and galactosan were analysed

together with radiocarbon 14C. The authors use their pragmatic coupled radiocarbon-LVG marker method (Salma et al. 2017) to attribute OC and EC to fossil fuel, biomass burning and biogenic origin fractions in Carpathian basin based on parallel, one week sampling period, at three sites in each season. Although the topic is important and data are new and valuable within the mentioned area, there are several major issues that should be answered before publishing the paper.

First, the naming of sampling periods (winter, spring, summer and autumn) suggests that the data are representative for these seasons. This is not true as sampling was performed during 14 (or even 7) consecutive days during each season and only 7 overlapping days were fully analysed for 14C. These sampling periods are too short to be representative for a season and therefore, months, when data were taken, are more proper for naming of the sampling periods. For the same reason, the authors should more concentrate on differences between the sites and less on "seasonal" characterisation and differences. More detailed weather characteristics for each sampling period can explain more various type of events that change differences among the sites.

Second, median values presented e.g. in the Table 2 or 4 can be representative for only part of the data especially if two types of atmospheric mixing were present during short sampling period. Therefore, either medians with high and low percentiles or averages with standard deviations should be presented together.

In addition, particle number concentrations paragraph (lines 437-442) is completely out of topic of the paper, it should be omitted together with related references.

Finally, the combination of OM/OC conversion factors used by authors is not logical and is not based on current scientific knowledge and must be corrected. Therefore, most of the calculations must be corrected.

Specific comments: Line 251-253 - The authors use conversion factor for city centre 1.6 and for suburban and rural cites 1.4. This is taken opposite way than it is usual. While both lower values 1.4 and used value 1.6 can be accepted for places with fresh traffic

aerosols – city centre the value 1.4 used for urban and rural background is unacceptably low. Some seasonal dependence of this factor can be also expected. Actually, cited work of Turpin and Lim 2001 says in its abstract: "This investigation suggests that 1.4 is the lowest reasonable estimate for the organic molecular weight per carbon weight for an urban aerosol and that 1.4 does not accurately represent the average organic molecular weight per carbon weight for a non-urban aerosol. Based on the current evaluation, ratios of 1.6 +/- 0.2 for urban aerosols and 2.1 +/- 0.2 for non-urban aerosols appear to be more accurate" Therefore, the calculation for suburban and rural cites have to be recalculated with higher conversion ratio OM/OC (at least also 1.6).

Specific comments Line 265 – It should stay "Their" instead of "They" Line 291 -293 – Measure of photochemical activity is not ozone concentration itself. Line 305 – WSOC vs SOA relation can be biased by biomass burning emissions. Therefore, the sentence needs correction. Compare also with lines 385 and 388. Line 378 - "minimum in summer" can be omitted as it is mentioned again in the next sentence. Lines 437-442 – It is out of topic, remove the paragraph Lines 456 – 457 – The sentence should be corrected, the results do not justify fully such sentence. Line 488 – OC/EC ratios can be influenced also by other effects, therefore less strong opinion would be more proper Line 547 – correlation coefficients are significant or insignificant based on given statistical criteria. Correct the sentence. Line 548 – "linear" relationship of OCFF with NO was seen for suburban site only (corr. coef. 0.93) while for city centre was only 0.39. Therefore, the sentence needs correction or clarification. Line 551 – the last sentence should be removed or corrected. The correlations can support results but not approve them. Line 569-570 – the differences in share of OCBIO are negligible in comparison with their uncertainty, therefore, no tendency can be retrieved from the data. Correct the sentence accordingly. Lines 662-664 - again OM/OC conversion factors – correct as mentioned above.

Graph 8 – if authors want to show differences in OC shares during their sampling periods they should stop call them seasonal differences, as their sampling periods

cannot fully represent seasons. Moreover, the lines in graphs are not representative for the data giving sometimes unrealistic impression about the data. Redo the graph

---

## Author Comment (AC1) · 1 Mar 2020

**Response to Referee number 1**

1st March 2020

The authors would like to thank Referee no. 1 sincerely for his/her very detailed, expertise and valuable comments to further improve and clarify the MS. We have considered all recommendations and made the appropriate alterations. We would like to touch upon that the Referee most likely evaluated the acp-2019-792-manuscript-version1.pdf and not the latest version acp-2019-792-manuscript-version2.pdf, which was created and resubmitted after the access review. In the latter version, some technical corrections and other smaller updates were adopted on the request of the co-editor, and a geographical map was included as well. This means that some issues raised by the Referee had been already handled and improved. Our specific responses are as follows, while the textual modifications can be followed in the marked-up version of the MS, which is attached.

**General comments**

English language should be thoroughly checked by a native speaker. Especially in the introduction some sentences are a bit hard to understand.

1. Several sentences of the MS, particularly in the Introduction, were shortened, simplified and the MS was also checked by a native English speaker. We also expect that the MS will receive English language copy-editing during the typesetting.

Chapter 2.1. It would help reader if you could name the stations in this chapter and clearly state stations called hereafter xx, zz and yy. Now there is many kind of variations of the names in the text. Maybe also include a more in-depth description of the area where stations are located (what kind of area, how many inhabitants, how much traffic/biomass burning/industry etc is in area, any prior knowledge about the expected sources?). Maybe the distance between stations or a map would help reader also. Please, add for all measured parameters the instrument, model and manufacturer. Also, it is bit hard to understand where e.g. DMPS was measuring and how long. Check and explain the used acronyms and terms. E.g. for elemental carbon both EC and soot are used, which can be very confusing to some of the readers. Also, terms carbonaceous aerosol, total carbon should be explained. More literature references should be added to the text to further discuss the results and their significance. Also more discussion about where these results could be utilized would be useful. The novelty value of results should be highlighted more!

2. The naming of the three measurement sites was unified all over the MS as regional background, suburban area and city centre. The suburban and centre sites are called together as urban locations. We also described the locations in more detail and added a geographical

map showing the three sampling sites and the distances between them. The description of the instruments was extended somewhat and clarified. The difference between soot and EC was made more exact (see also Responses no. 4 and 5). The total carbon was already defined as TC=OC+EC in the line 85 of the MS ver. 1. We put more emphasize on the novelty of the approach and results, and on their utilization and potentials in Sect. 5, Conclusions.

Tables 2,3: The sampling periods for all stations are different. At the Central station the sampling period is much shorter. Are the mean/meadian values and ratios calculated for all the samples or only for the seven simultaneously collected samples? If sampling times are not exactly same, is it fair to compare the results of stations? e.g. some episode could change the concentrations and affect the observed mean values significantly.. if this episode is only included in longer timeseries measured in Background/suburban areas, this could affect the comparison when the results of different stations are compared.

3.  The average atmospheric concentrations in Tables 2 and S2 were calculated by considering all sampling days. The major reason for doing this is to characterize the environments and months as representatively as possible. These overall mean values were compared to previous results. This is mentioned explicitly in the text now. We also added a brief note on constrains imposed by not completely overlapping sampling intervals at the city centre with respect to the other two sites. Of the three locations, we have the most extensive information for the city centre, where the present median concentrations were perfectly in line with the earlier results. Atmospheric concentrations in the Carpathian basin are often governed by local meteorology (Salma et al.: Elemental and organic carbon in urban canyon and background environments in Budapest, Hungary, Atmos. Environ., 38, 27, 2004). Since the meteorological data during the sampling intervals were in line with monthly characteristics and without any extremes, it can be expected that the aerosol samples collected represent the months correctly. This confirms the comparison. We also modified the naming of the sampling intervals in the entire MS to express that an interval is more related to a month than to a whole season. A note was also added on the representativity of the sampling intervals into Sect. 3. The radiocarbon analysis was performed on seven selected samples which were collected in parallel on fully overlapping days at each sampling location and in each month. The comparison of the apportioned concentrations, their contributions and their interpretations (which are the principal objectives of this paper) are fully justified. All these arguments were now included briefly into the text at several places.

**Minor comments**

4. The term "carbonaceous aerosol constituents/species" refers to the carbonaceous components of the particles. In the fine size fraction, they ordinarily represent organic compounds and soot. This naming is quite straightforward, it is explained in basic textbooks (e.g. Gelencsér: Carbonaceous aerosol, Springer, 2004, p. 2) and is generally accepted. The carbonaceous species usually occur together with inorganic compounds, and, therefore, the term "carbonaceous particles" is – strictly speaking – already a fiction in most cases because it expresses the carbonaceous constituents of the aerosol particles only. We prefer to use the former and well-defined term in all our papers. We added a reference on this issue (see also answer no. 5).

Line 39; define soot (as there is also EC)

5. The meaning of soot was explained as requested. In addition, soot was also related to EC at the first occurrence of the EC. We also added a reference dealing with this complex terminological issue in detail.

Line s43-51: Sentences are bit long and hard to read. Clarify this and maybe specify if these consequences in the list are positive or negative in the nature.

6. The sentences were shortened and simplified. Discussions of the effects listed, however, would be rather complex and long, and more importantly, they are not in line with the purpose of the MS.

Line 52: "Fuel wood"? does this refer to biomass combustion in residential scale?

7. Fuel wood is made from trunk and larger branches of trees and are burnt in both residential and industrial appliances. The sentence was extended.

Line 59-61: "Huge number, composite character, spatial and temporal variability of the sources together with the complex mixture and atmospheric transformation of their products make the quantification of these source types or their inventory-based source assessment challenging" Clarify this sentence, it is bit hard to understand.

8. We tried to reformulate the sentence without leaving out any important process.

Line 62-63: "There are several methods to apportion the particulate matter (PM) mass or carbonaceous species among some or all major source types." please clarify this sentence

9.  The sentence was clarified.

Lines 63-69. Sentence is really long, maybe split to 2-3 shorter sentences?

10. The sentence was split into two separate parts. The first sentence of them may still seem somewhat longer but – as a matter of fact – it is a list of available methods, which we would like to keep as one item.

Line 74: "The latter molecule is often applied together with its stereoisomers mannosan (MAN) and galactosan (GAN) since.." maybe change to " Monosaccharide anhydride analysis often contains stereoisomers mannosan (MAN) and galactosan (GAN) in addition to levoglucosan since..

11. Adopted.

Line 100: what is the "latter type" referring to? please clarify

12. The expression "latter type" referred to biogenic sources. The sentence was modified.

Table 1. Why there is extra space between date and month as well as between the month and year in all time periods? please check the journal instructions how to give the dates..

13. The writing of the dates and times was adjusted the journal instructions in the whole MS.

Lines 152-167: Please add the model and manufacturer for all the instruments, provide the instrument information for the meteorological data as well information in which stations these instruments were used. E.g. was DMPS run in all stations constantly, or was one dmps moved between the stations? Maybe a table with station, instruments, models and measured components would help reader to understand the situation.

14. The parts on particle number concentrations and the related reference were removed on the request of the Referee no. 2. We included the types and manufacturers of the meteorological instruments which make it feasible to obtain further information on them. We consider listing all details for measuring auxiliary variables, which are part of a quality-controlled national meteorological network not completely justifiable since these long and detailed lists would make the article over-descriptive and would detract the attention from the aimed messages.

Line 169: Add balance model and manufacturer

15. The requested information was added.

Line 173: where does this LOQ value for PM mass comes from?

16. The specified limit was determined from evaluating measured data for several box blank filters identical to our substrate within a separate exercise. The procedure included for instance uncertainties of weighing, sampled air volume determination and environmental conditions according to EN12341:2014. We added a brief note on this into the text.

Line 198: what is origin of the LVG observed in the blank filters? please add how much levoglucosan was observed in the blanks.. has this kind of blank values seen in other studies?

17. The LVG amount in the blank filters can be related to the sampling itself, to chemicals used, to various chemical and sample preparation procedures performed and to the variations of the baseline of the measurement. We added a reference on them (Maenhaut et al.: Assessment of the contribution from wood burning to the $PM_{10}$ aerosol in Flanders, Belgium, Sci. Total Environ., 437, 226, 2012) , which showed similar blank values. The blanks were the largest in comparison to the corrected values in the summer samples, in which the measured LVG amounts were approximately ten times larger than in the blank filters. In all the other samples, the relative contributions of the blanks were substantially smaller than this. We added a related brief comment to the text.

Line 203: which days?

18. The overlapping days were given in an explicit way.

Line 236-240: "Whenever it was possible, the comparisons of atmospheric concentration, other variables or their ratios with respect to sites or seasons were accomplished by calculating first the ratios on a sample-by-sample or day-by-day basis and then by averaging these individual ratios for the subset under consideration". Please explain what variables/ratios this refers to?

19. A sentence was modified to be more specific.

Line 275: Maybe add some values for average temperature, wind etc meteorological parameters to article also (not only supplement) as people not living in Budabest may not know the normal local conditions mean.

20. The average values for the complete list of meteorological variables are given in Table S5 in the Supplement. We would like to avoid showing only some selected variables separately, and instead, we further emphasized that they are fully available in the Supplement.

Line 284-286: "The former variable represents the bulk fine PM; EC is a typical primary aerosol constituent, while WSOC is expresses the SOA." Sentence is hard to understand, please clarify what this means.

21. A typo unfortunately remained in the sentence which made it difficult to understand. It was corrected and in addition, the sentence was further clarified.

Chapters 3.2-3.6 please add some numberical values to text also. Would be also useful to compare more to literature wether the values were as expected or may be lower/higher..

22. We compared our present results better to existing partial or overlapping information and discussed the consequences of the inter-comparison. We also added some further references.

Line 391-395: maybe this information should be in experimental chapter?

23. The suggestion could indeed be a plausible option. Our specific intention by keeping this short but important discussion among the results was to attract more attention to possible contamination by anthropogenic $^{14}$C in large cities. This risk is not very often mentioned in urban radiocarbon studies.

Line 359-365; "These can be explained by larger intensity of soot emissions from incomplete burning (road vehicles, residential heating and cooking by solid fuel), which is a typical anthropogenic source, and which is associated with seasonal variation (e.g. due to residential heating) as well as with constant sources (e.g. due to traffic or cooking) over a year." what is the seasonal variation observed for road vehicles? how they differ from traffic that does not have seasonal variation?

24. The seasonal variability was related to residential heating, while the road traffic is expected to be a source with more-or-less constant daily average intensity of soot particles (see Salma et al., Elemental and organic carbon in urban canyon and background environments in Budapest, Hungary, Atmos. Environ., 38, 27, 2004). The related sentence was reformulated to avoid the possible misunderstanding.

Line 501-502: Did temperature have similar trend as OCbio? I would have expected to see highest OCbio concentrations in summer.. or is there a reason why autumn OCbio was higher? how does this observation compare to other studies?

25. The tendency in the air temperature can be seen in Table S5 in the Supplement. It did not change in line with the $OC_{BIO}$ (in contrast to species apportioned to BB). Their coefficients of correlations at all sampling sites are summarized in Table S6, which indicate a modest

(linear) relationship between $T$ and $OC_{BIO}$ in the regional background and insignificant dependencies at the urban sites. The links between them are rather complex and not fully uncovered. The $OC_{BIO}$ depends on many other atmospheric properties and parameters than $T$ in a multifactorial manner. This set including for instance the changes of the SOA yield for mixed air masses of biogenic and urban origin. Our conclusion is in line with a recent and very important article dealing with the SOA formation (McFiggans et al.: Secondary organic aerosol reduced by mixture of atmospheric vapours, Nature, 565, 587, 2019). It is needed to continue the studies on the SOA yield in these specific atmospheric environments by dedicated experiments and methods to get better insights into and explanations for our apportioned results. The related text was reformulated to further emphasize these aspects, and a brief comparison was also added.

Chapter 3.6 and conclusions: the impact of results is now discussed from the air quality point of view. Maybe add something about climate and health point of view also? assumably the anthropogenic emissions and BC have different, likely more negative health impacts. BC has also strong climate impact..

26. The consequences of the study on the air quality in the Carpathian Basin are dealt with in a separate section (3.6). These are related to health effects as well. We amended several modifications into the Conclusions section to indicate their climate implications as well.

Imre Salma
corresponding author

---

## Author Comment (AC2)

**Response to Referee number 2**

The authors would like to thank Referee no. 2 very much for his/her very detailed, expertise and valuable comments to further improve and clarify the MS. We have considered all recommendations and made the appropriate alterations. Our specific responses are as follows, while the textual modifications were amended and can be followed in the marked-up version of the MS, which is attached.

**General comments**

First, the naming of sampling periods (winter, spring, summer and autumn) suggests that the data are representative for these seasons. This is not true as sampling was performed during 14 (or even 7) consecutive days during each season and only 7 over-lapping days were fully analysed for 14C. These sampling periods are too short to be representative for a season and therefore, months, when data were taken, are more proper for naming of the sampling periods. For the same reason, the authors should more concentrate on differences between the sites and less on "seasonal" characterisation and differences. More detailed weather characteristics for each sampling period can explain more various type of events that change differences among the sites.

1. We modified the naming of the sampling periods in the entire MS to express that they are related more to a month than to a whole season. A note was also added on the representativity of the sampling intervals into Sect. 3 to clarify the situation more carefully and considerately. We included the aspect raised by the Referee in the second part of this comment into the interpretations of the data, and performed several modifications of the text accordingly.

Second, median values presented e.g. in the Table 2 or 4 can be representative for only part of the data especially if two types of atmospheric mixing were present during short sampling period. Therefore, either medians with high and low percentiles or averages with standard deviations should be presented together. In addition, particle number concentrations paragraph (lines 437-442) is completely out of topic of the paper, it should be omitted together with related references.

2. We added new tables into the Supplement (Tables S2 and S4), which contain the means and SDs of atmospheric concentrations of aerosol constituents and gases, and explained its motivation in the MS. We originally included the particle number concentrations to demonstrate the decoupling between PM mass and particle number. As requested, we removed the related paragraphs and reference.

Finally, the combination of OM/OC conversion factors used by authors is not logical and is not based on current scientific knowledge and must be corrected. Therefore, most of the calculations must be corrected. Line 251-253 - The authors use conversion factor for city centre 1.6 and for suburban and rural cites 1.4. This is taken opposite way than it is usual. While both lower values 1.4 and used value 1.6 can be accepted for places with fresh traffic aerosols – city centre the value 1.4 used for urban and rural background is unacceptably low. Some seasonal dependence of this factor can be also expected. Actually, cited work of Turpin and Lim 2001 says in its abstract: "This investigation suggests that 1.4 is the lowest reasonable estimate for the organic molecular weight per carbon weight for a non-urban aerosol. Based on the current evaluation, ratios of 1.6 +/- 0.2 for urban aerosols appear to be more accurate" Therefore, the calculation for suburban and rural cites have to be recalculated with higher conversion ratio OM/OC (at least also 1.6).

3. The organic aerosol-to-organic carbon (OC) mass conversion factor is an estimate of the average molecular mass per C atom for organic matter (OM) in general. It is site-dependent and can have seasonal and diurnal variations as well. Therefore, the factor cannot be considered as a conclusive or constant/generally valid value. It is usually derived by indirect considerations (Russell, Aerosol organic-mass-to-organic-carbon ratio measurements, Environ. Sci. Technol., 37, 2982, 2003). Mass conversion factors between 1.2 and 1.4 were estimated for fine atmospheric aerosol in mildly oxidizing atmospheric environments (Turpin et al.: Measuring and simulating particulate organics in the atmosphere: problems and prospects, Atmos. Environ., 34, 2983, 2000). Some further studies suggest that a factor of 1.6±0.2 describes better the oxidizing urban environments (Turpin and Lim: Species contributions to PM2.5 mass concentrations: revisiting common assumptions for estimating organic mass, Aerosol Sci. Technol., 35, 602, 2001). Identical partial mass conversion factors of 1.81 were obtained for HULIS both at a rural site of the Carpathian Basin and in Budapest (Kiss et al.: Characterization of water-soluble organic matter isolated from atmospheric fine aerosol, J. Geophys. Res., 107(D21), 8339, 2002; Salma et al., Sampling artefacts, concentration and chemical composition of fine water-soluble organic carbon and humic-like substances in a continental urban atmospheric environment, Atmos. Environ., 41, 4106, 2007, respectively). HULIS are comprised primarily of a complex multi-component mixture of compounds that bear aliphatic chains with carboxyl, hydroxyl, carbonyl or phenol terminal groups. Thus, they contain relatively rather large number of heteroatoms to C but exhibit an OM/OC ratio of "only" 1.81, while they mass contribution to OC could be 20–30%. It should also be noted that the conversion factor is one of the most substantial sources of uncertainty in aerosol chemical mass closure calculations involving OM. It was estimated that the

relative uncertainty associated with the conversion is approximately 30% (Maenhaut et al., Assessment of the contribution from wood burning to the  $PM_{10}$  aerosol in Flanders, Belgium, Sci. Total Environ., 437, 226, 2012). In the present study, we adopted the factor of 1.4 for the regional and suburban environments and the factor of 1.6 for the city centre. We would like to keep our selection because of several reasons. 1) We think that the larger factor mentioned and quoted by the Referee for rural and suburban environments is primarily valid for chemically aged aerosol, which was not the typical case at our sampling sites in the Carpathian Basin. Most aerosol particles are generated by local or regional sources here. 2) The two factors of 1.4 and 1.6 under discussion have uncertainties which are identical to or even larger than the differences between the factors. Moreover, the factor does not affect at all the major objectives of the MS, namely the apportionment of the basic classes of OC and EC from FF combustion, BB and biogenic sources, and their contributions to TC. 3) Our previous studied in this field and geographical area justify our selection since we obtain consistency in the results in general for various organic aerosol types and environmental types within the Carpathian Basin. 4) We utilized the present ratios in our several earlier publications (including ACP articles as well) and keeping the present conversion factors also facilitates the comparison among the present and previous results. As a compromise, we extended the related parts of the MS with these discussions and explanations, and further emphasised the role of methodological uncertainties or limitations in the text whenever it was relevant.

**Specific comments**

Line 265 - It should stay "Their" instead of "They"

4. Corrected.

Line 291-293 - Measure of photochemical activity is not ozone concentration itself

5. The sentence was reformulated.

Line 305 – WSOC vs SOA relation can be biased by biomass burning emissions. Therefore, the sentence needs correction. Compare also with lines 385 and 388.

6. The sentence was extended by this aspect as well.

– 3 –

Line 378 - "minimum in summer" can be omitted as it is mentioned again in the next sentence.

7. The expression was removed.

Lines 437-442 – It is out of topic, remove the paragraph

8. We originally included the information on the particle number concentrations to demonstrate the decoupling between PM mass and particle number. Nevertheless, we removed the paragraph as requested.

Lines 456 - 457 – The sentence should be corrected, the results do not justify fully such sentence.

9. The related part of the sentence was deleted.

Line 488 – OC/EC ratios can be influenced also by other effects, therefore less strong opinion would be more proper 10. The sentence was reformulated in the requested manner.

Line 547 – correlation coefficients are significant or insignificant based on given statistical criteria. Correct the sentence.

11. The sentence was extended by the significance limit.

Line 548 – "linear" relationship of OC\_FF with NO was seen for suburban site only (corr. coef. 0.93) while for city centre was only 0.39. Therefore, the sentence needs correction or clarification.

12. The sentence was corrected and extended into a more precise and clearer formulation.

Line 551 – the last sentence should be removed or corrected. The correlations can support results but not approve them.

13. The sentence was removed.

Line 569-570 – the differences in share of OC\_BIO are negligible in comparison with their uncertainty, therefore, no tendency can be retrieved from the data. Correct the sentence accordingly.

14. The sentence was changed to include this limitation.

Lines 662-664 - again OM/OC conversion factors - correct as mentioned above.

15. Section 3.6 deals with the potentials of the apportioned chemical species on the air quality as it is explicitly expressed in the text, and some rough assumptions, which are also outlined, were utilized. From this aspect, the differences caused by the two possible OM/OC conversion factors of 1.4 or 1.6 seem unimportant. The limits of the approach were further explained and discussed in Sect. 3.6. See also the answer no. 3.

Graph 8 - if authors want to show differences in OC shares during their sampling periods they should stop call them seasonal differences, as their sampling periods cannot fully represent seasons. Moreover, the lines in graphs are not representative for the data giving sometimes unrealistic impression about the data. Redo the graph.

16. The naming of the sampling periods was modified as requested and the name of the corresponding months were adopted instead. The line with a time tendency in question was removed from the plot.

Imre Salma corresponding author